# Autochtonous Strain *Enterococcus faecium* EF2019(CCM7420), Its Bacteriocin and Their Beneficial Effects in Broiler Rabbits—A Review

**DOI:** 10.3390/ani10071188

**Published:** 2020-07-14

**Authors:** Monika Pogány Simonová, Ľubica Chrastinová, Andrea Lauková

**Affiliations:** 1Institute of Animal Physiology, Centre of Biosciences of Slovak Academy of Sciences, Šoltésovej 4-6, 04001 Kosice, Slovakia; 2Institute for Nutrition, National Agricultural and Food Centre, Hlohovecká 2, 951 41 Nitra-Lužianky, Slovakia; lubica.chrastinova@nppc.sk

**Keywords:** rabbit, *Enterococcus faecium*, enterocin, microbiota, intestinal morphology, phagocytic activity, serum biochemistry, meat quality, weight gain

## Abstract

**Simple Summary:**

Weaning is the most important and critical period in rabbits breeding; the cecal digestion is very complex and only small dietary and/or environmental changes can disturb the stable microbial population/fermentation and gut health, leading to digestive dysbiosis and increased morbidity, often with fatal outcome and big economic losses. Control of the microbiota, prevention of digestive disturbances and improving gut health and immunity can be achieved through the natural substances application in rabbit nutrition. While probiotics are frequently used in rabbit farms, the in vivo administration of bacteriocins (antimicrobial substances produced by bacteria, which usually also possess probiotic properties) in these animals is often limited and has become an area of research activity. Moreover, the most of probiotic strains used in rabbits are non-autochthonous (have a different origin than the rabbits ecosystem). Therefore, our study focused on improving rabbits’ health using the autochthonous strain *Enterococcus faecium* EF2019 (CCM7420) and its enterocin (Ent7420) in broiler rabbits. The antibacterial and anticoccidial effect of additives was observed, with good colonization ability of the CCM7420 strain. Both additives showed a tendency to modulate the serum biochemistry parameters and to improve the immunity, jejunal morphology, weight gains, feed conversion ratio and meat quality (physicochemical traits and mineral content).

**Abstract:**

The present review evaluates and compares the effects achieved after application of rabbit-derived bacteriocin-producing strain *Enterococcus faecium* CCM7420 with probiotic properties and its bacteriocin Ent7420. The experiments included varying duration of application (14 and 21 days), form of application (fresh culture and lyophilized form), combination with herbal extract and application of the partially purified enterocin—Ent7420, produced by this strain. Results from these studies showed that *E. faecium* CCM7420 strain was able to colonize the gastrointestinal tract (caecum) of rabbits (in the range < 1.0–6.7 log cycle, respectively 3.66 log cycle on average), to change the composition of intestinal microbiota (increased lactic acid bacteria, reduced counts of coliforms, clostridia and staphylococci), to modulate the immunity (significant increase of phagocytic activity), morphometry (enlargement absorption surface in jejunum, higher villi height:crypt depth (VH:CD) ratio), physiological (serum biochemistry; altered total proteins, glucose and triglycerides levels) and parasitological (*Eimeria* sp. oocysts) parameters and to improve weight gains (in the range 4.8–22.0%, respectively 11.2% on average), feed conversion ratio and meat quality (physicochemical traits and mineral content).

## 1. Introduction

Rabbit breeding has a great potential because of the small body size, short generation interval, rapid growth rate, high productive capacity and healthy, easily digestible meat of rabbits [1,2]. Moreover, rabbits can convert a higher amount (20%) of the protein they eat into edible meat, compared with pigs (16–18%) and cattle (8–12%; [3]). In several European countries, in which the rabbit breeding has a long history and high production efficiency, presently is regressing, whereas in the developing countries of the world rabbit farming has become to an important emerging enterprise. In growing rabbits, the most critical period is the weaning, when the kits are separated from mother and the milk is substituted with solid feed [4]. During these environmental and physiological changes/stresses, the rabbits are very sensitive to digestive disturbances, also called non specific enteritis (usually caused by dietary stresses, parasites—*Coccidia* and bacteria—*Clostridia* sp. and enteropathogenic *Escherichia coli*) and gastrointestinal infections (epizootic rabbit enteropathy—ERE, a multifactorial gastrointestinal syndrome, [5]). These dietary and bacterial changes are the main reason of morbidity and mortality and have negative effects on feed consumption, growth performance and health status of animals in this period and also on the economic aspects of rabbit farming.

To overcome this period, to reduce economic losses and to improve and stabilize the health status and gastrointestinal tract development, antibiotic growth promoters (AGPs) have been widely used for years. Although, these synthetic drugs showed good effects on production indicators, on the other hand, there was a risk of increasing resistance to antibiotics and transferring of antibiotic resistance genes from animal to human, which also threatened the human health and quality of meat and food [6]. For this reason, AGPs have been banned by the European Union (began in 1986 in Sweden and completely banned in January 2006, when the last four antibiotics have been permitted as feed additives was no longer allowed to be marketed or used from this date; IP/03/1058; [7,8]). As a result of the ban, researchers had to substitute AGPs and to find new feed additives that were supposed to be safer, without leaving residues and spreading resistance to themselves, but also improving health and productivity of rabbits. Therefore, the antibiotics have been replaced with new, naturally based supplements: probiotics, prebiotics, synbiotics, enzymes, bacteriocins, organic acids, herbs and their extracts, which are well-tried tools for disease prevention and therapy in various animal species, including rabbits [9].

The last two decades have seen a substantial increase in the use natural supplements and/or additives in animal nutrition, in which their antimicrobial activity has been highlighted many times. The EU in its Regulation EC 1831/2003 defined the terms “feed additives” as “substances, microorganisms or preparations, other than feed material and premixtures, which are intentionally added to feed or water” and the “antimicrobials” as “substances produced either synthetically or naturally, used to kill or inhibit the growth of microorganisms, including bacteria, viruses or fungi, or of parasites, in particular protozoa” [10]. According to the World Health Organization (WHO) probiotics are defined as “live microorganisms which, when administered in adequate amount, confer a health benefit on the host”. Although this definition is widely accepted, a 2007 guidance document from the European Commission on Regulation EC 1924/2006 on nutrition and health claims (NHCR) categorizes the term ‘probiotic’ as a health claim on the basis that it implies a health benefit. For this reason, the term “beneficial microbes” is used more often instead of a “probiotic microorganism” [11].

The use of several natural feed additives has already been reviewed in rabbit breeding [10,12,13,14,15]. The positive effects of probiotics and their antibacterial products—bacteriocins on health, growth performance, nutrient utilization and metabolism changes, microbial composition [14,16,17,18,19,20,21,22,23,24,25,26,27,28,29,30,31,32,33,34,35], blood serum biochemistry, oxidative stress, immune response, intestinal morphology [21,24,28,29,30,31,34,36,37,38] and meat quality of rabbits [32,34,39,40,41,42,43] was described. However, in spite of the achieved results, there are still few declared probiotic preparations based (Lactina, Toyocerin; [44,45]; Prorabbit—declared in Slovakia; [25]) and detailed studies of microorganisms with beneficial properties that have also the ability to produce antimicrobial substances, enterocins.

The aim of this review was to summarize all achieved properties and physiological effects of the bacteriocin-producing strain with probiotic properties *Enterococcus faecium* CCM7420 (EF2019 previous working labeling, [46]) isolated in 2003 from rabbit feces in the Laboratory of Animal Microbiology of the Institute of Animal Physiology, Centre of Biosciences of the Slovak Academy of Sciences (Košice, Slovakia) and tested to date in 180 rabbits. These experiments included varying duration of application (2 and 3 weeks), form of application (fresh culture in water; the concentration of cells was ×10^9^ CFU/mL in a dose 500 μL/animal/day; lyophilized (freeze-dried) form rehydrated in water (×10^9^ CFU/mL; dose 500 μL/animal/day) as well as mixed in feed and pelleted (15 g/100 kg feed), application of its partially purified bacteriocin (PPB)—enterocin (Ent) EF2019 (applied into water) and fresh culture in combination with natural substance (*Eleutherococcus senticosus*).

## 2. *Enterococcus faecium* CCM7420 (EF2019) and Its Bacteriocin-Enterocin (Ent7420)

*Enterococcus faecium* EF2019 (CCM7420) is a bacteriocin-producing strain [47], which was isolated from the rabbit feces and genetically confirmed by the PCR method and subsequently by MALDI-TOF mass spectrophotometry as well as the sequencing procedure of this strain was provided (Dr. Kopčáková, IAP CBs SAS). This strain produces lactic acid, tolerates low pH (3.0; 63% surviving of cells) and is able to grow even in 5% oxgall—bile (80% surviving of cells), shows sensitivity to antibiotics, including vancomycin [25,48] and possess lipolytic activity [49]. Other unpublished data suggests that the CCM7420 does not produce biogenic amines and enzymes such as β-glucuronidase, β-galactosidase or N-acetyl-β-glucosaminidase (enzymes produced by unfriendly gut bacteria; their increased levels are usually the indicators of colon cancer), and it does not show any gelatinase (absence of the *gelE* gene) or hemolytic activities with low ability to form biofilm (0.092). The strain was deponed into Czech Collection of Microorganisms in Brno, Czech Republic to have number CCM7420. This strain showed the broadest inhibitory activity from all tested rabbits enterococcal strains against the indicators *E. avium* EA5, *Listeria innocua* LMG13568 and *L. monocytogenes* CCM4699 and against other tested enterococci and staphylococci tested such as clostridia, pseudomonads, enterobacteria and coliform bacteria [48]. The presence of the structural genes for enterocins (ent) A, P and L50B was detected; however, the CCM7420 did not possessed gene for ent B [47]. The molecular mass of its bacteriocin-like substance ranged from 3 to 10 kDa. Proteinaceous substance produced by CCM7420 strain was partially purified (partially purified bacteriocin (PPB) or enterocin (Ent) 2019 =7420). It is thermostable substance as well as stable at pH 4.0, 7.0 and 9.0. Its production starts in early logarithmic growth phase and it culminates in the late logarithmic phase of CCM7420 strain growth. By its properties, it can probably be included in the II. classification group of bacteriocins. Ent2019 or Ent7420 added to the growing strain *L. innocua* LMG13568 (after 4 h) inhibited its growth already at 1 h after enterocin addition with a difference of 1.5 log cycles (5 h of cultivation). This effect was prolonged up to 24 h. The Ent7420 was tested against more than 300 strains of enterococci, staphylococci, clostridia, pseudomonads, enterobacteria and coliforms [48]. The inhibitory activity of this substance was preserved after 24 months of storage at −20 °C (6400 AU/mL; [48]) and also after lyophilization (freeze dried) and redissolution in PBS buffer (25600 AU/mL; not published data). The CCM7420 strain is currently available in the ProRabbit, probiotic product for rabbits and other rodents, made by the International Probiotic Company Košice (Slovakia).

## 3. Application Effects of *E. faecium* CCM7420 and Its Enterocin Ent7420 Observed in Experiments

### 3.1. Effect on Growth Performance

The results from all these experiments indicated that the CCM7420 strain could improve the average daily weight gain (ADWG; between 4.8 and 22.0%; Table 1) regardless of the form of bacteriocin-producing strain (fresh, *p* < 0.001 compared to control data; or freeze dried-lyophilized—numerical increase) and its application time (2 or 3 weeks). The feed conversion ratio (FCR) was influenced only through the Ent7420 administration and the combinative application of the CCM7420 strain with *Eleutherococcus senticosus* extract. The effect of probiotics, including registered probiotic preparations and new beneficial microorganisms on the growth performance of rabbits have been already described/reviewed; they usually confirmed the increased body weight [14,15,17,19,24,25,31,32]. The most of these studies described faster growth and higher weight gain of rabbits using probiotic preparations and feed additives based on the following bacterial strains and yeasts alone or in their combinations: *Saccharomyces cerevisiae*, *S. boulardii*, *Bacillus licheniformis*, *B. cereus*, *B. cereus* var. *toyoi*, *Pediococcus acidilactici*, *Lactococcus lactis*, *Lactobacillus acidophilus*, *L. plantarum*, *L. helveticus*, *L. delbrueckii* and *L. sporogenes* [14]. Up to now, only several commercial products recommended especially for rabbits contain especially the species strain *Enterococcus faecium* as a component of the probiotic bacterial mix (Lactina—*E. faecium* NBIMCC 8270 [44]; Prorabbit—*E. faecium* CCM7420 [25]) or for companion animals with diarrhea (Pro-enteric Triplex—*E. faecium* DSM 10663/NCIMB 10415 [50]). In all experiments, higher ADWG was noted during CCM7420 administration. Surprisingly, the highest increase of ADWG (by 22%, Table 1) was noted after 2 weeks addition (model experiment, only seven animals were used in the group). The three weeks long CCM7420 dietary inclusion also improved the growth performance; however, only by 6.7% on average (from results of fresh culture application in two experiments; [25,26]). Similarly to our results, Lauková et al. [21] and Szabóová et al. [35] also reported higher ADWG through bacteriocinogenic and probiotic *E. faecium* AL41 (CCM8558) and CCM4231 strains application in rabbits. Improved body weight in rabbits was also noted after probiotics administration by Bovera et al. [30], Bhatt et al. [32] and Kalma et al. [51]. On the other hand, after PPB CCM7420 addition increased the body weight only slightly (by 2.2%, Table 2), but in this group, better FCR was achieved, compared to strain application. The bacteriocin dietary inclusion showed better feed conversion also in the case of other bacteriocins: EntM, EntCCM4231, EntEF55 and gallidermin [21,52,53,54], applied in rabbit husbandry.

### 3.2. Effect on Fecal Microbiota

In all experiments, administration of *E. faecium* CCM7420 was associated with increased enterococci (*p* < 0.01) and lactic acid bacteria (LAB) counts during the treatment period by a 1.3 log cycle on average (Table 2), while the application form or time length had no impact on the bacterial counts increase. The numerical increase of enterococci and LAB was recorded also in previous experiments with non-autochthonous *E. faecium* probiotic strains inclusion in rabbits [21,29,56]. Other authors also noted abundance of microflora in caecum and higher lactobacilli counts after *Lactobacillus* strains application [57]. Outgoing from results of several studies focusing on the molecular profiling of rabbits gut [58,59,60] and colonization ability of applied enterococcal probiotic strains [33], enterococci were found as predominant microbiota from the phyla *Firmicutes* and they were able also to colonize the caecum and the small intestine (ileum and jejunum) in sufficient counts. The microbiological examinations in fecal samples confirmed the presence of *E. faecium* CCM7420 strain. This strain was able to colonize the digestive tract of rabbits, reaching counts in the range 2.8–6.7 log cycle during the 2 or 3 weeks treatment. These numbers are comparable with the level of autochthonous probiotic strains of several *Enterococcus* spp. after their application in rabbits [35]. Obviously, decreased CCM7420 counts (1.0–3.3 log cycle) were noted in the post-treatment period (3 weeks after strain cessation), but it was still able to persist in the rabbit’s intestine. However, the lowest counts in fecal samples were achieved during the experimental application of the lyophilized CCM7420 strain [60], this level is comparable to that one achieved through fresh culture addition of non-autochthonous, bacteriocinogenic and the probiotic *E. faecium* AL41 (CCM8558) strain in rabbits [21]. Important results concerning the spoilage microbiota, including clostridia, coliforms and staphylococci were described in our experiments. Significant reduction of coagulase-positive staphylococci was noted in fecal samples of rabbits, administering fresh culture of CCM7420 (*p* < 0.01; [24]); the *S. aureus* counts were also reduced by 0.5–1.2 log; *p* < 0.001 [25,26]. The CCM7420 strain seems to be useful in rabbits suffering from diarrhea disturbances involving coliforms (reduced by 1.6 log; *p* < 0.001 [26]) and/or clostridia (reduced by 0.5–1.5 log; *p* < 0.05 [25,26]). Similarly to our results, decreased counts of coliforms, *S. aureus* and clostridia were presented in probiotic-treated rabbits [21,23,29]. In rabbits administering the enterocin Ent7420, reduction in coliforms, coagulase-positive staphylococci, including *S. aureus* was noted [25]. Our results are in accordance to those presented by Lauková et al. [21], who tested the effect of nisin on the rabbits gut microbiota and noted reduced counts of the most bacterial species. The gut microbiota is an important constituent in the intestine’s mucosal barrier; the increase of the host defense had been already demonstrated by the application of potentially beneficial microorganisms and other natural antimicrobials (bacteriocins, organic acids and plant extracts; [61]).

Changes in bacterial composition—decrease of fecal coliforms, *Pseudomonas*-like sp., *Clostridium*-like sp. and *S. aureus*—during the additives supplementation confirm the antibacterial effect of CCM7420 strain. The inhibitory effect of the bacteriocin-producing and probiotic enterococci and their enterocins on rabbits’ intestinal microbiota was already reported in our previous studies [24,25,28,29,52]. Kritas et al. [23] also described the lower frequency of *E. coli* and *C. perfringens* in rabbits treated by probiotic. The dominancy in antimicrobial activity of CCM7420 strain combined with *E. senticosus* can be confirmed by the fact that the *E. senticosus* extract possesses slight or no antimicrobial activity [26]. We supposed that the dietary modulation of the gastrointestinal microbiota by natural antimicrobial substances could result in an enhancement of colonization resistance against potentially pathogenic bacteria.

### 3.3. Effect on Eimeria sp. Oocysts

Coccidiosis is one of the most frequent and prevalent parasitic diseases in rabbit farms. The most markedly and typical symptoms are weight loss, diarrhea (from mild intermittent to severe) with the presence of blood and/or mucus in feces, which leads, through dehydration, to the mortality of animals. The high morbidity and mortality rates among all ages, especially in the young rabbits may be responsible for important economic losses [62]. The oocysts are always present in the intestines of rabbits and they cannot be completely eliminated even by the use of coccidiostat because of the caecotrophy and the symptomless, but potential source of infections of adults. These oocysts are able to cause not only pure eimeriosis after their multiplication and massive infection, but they may also be the cause of multifactorial diseases, when associated with other bacterial or viral infections. Eimeria infections can cause severe disease depending on *Eimeria* species, especially in young animals and the highest incidence of oocysts was usually found around the weaning period [63]. The EU has banned the use of antibiotics as feed additives for growth promotion in animals since 2005 [64]. Today, these antibiotics are replaced with alternative anticoccidials, including prebiotics and probiotics, based on their bactericidal and/or bacteriostatic activities, with immunostimulation and improved growth performance and productivity of the host organism. The experimental application of the CCM7420 strain and its Ent7420 was associated with the reduction of fecal *Eimeria* sp. oocysts. The different treatment period had no impact on oocysts reduction. When the fresh culture of the CCM7420 strain at concentration ×10^9^ CFU/mL/g was applied only for 2 weeks to compare its effect with the commonly used probiotic strain *Lactobacillus rhamnosus* GG, the decrease in oocysts counts was observed after probiotic application (8.3 × 10^1^ OPG; oocysts per 1 g of feces) compared to control data (1.5 × 10^4^ OPG), but also compared to the initial counts in experimental group (7.5 × 10^2^ OPG). This reduction effect was maintained until the end of the experiment, also after cessation of the CCM7420 strain. Moreover, at the end of the experiment (at day 42), the difference one order of magnitude in *Eimeria* sp. oocysts was found comparing the control (1.5 × 10^4^ OPG) and the experimental groups (7.2 × 10^3^ OPG; [26]). On the other hand, when CCM7420 was applied with its bacteriocin Ent7420 in rabbits through 3 weeks, after 1 week of their addition, oocysts showed a trend towards a numerical reduction (not significant; [25]) in both experimental groups compared to the control group (Table 1). Surprisingly, at the end of CCM7420 strain addition, increased oocyst counts was observed in this group, while Ent7420 administration was more effective due to a significant reduction in oocysts (*p* < 0.05) at the end of probiotic application (day 21). This finding could be explained by the irregular excretion of oocysts. In the group with Ent7420 addition, a decreasing tendency of oocysts occurrence (not significant) was observed up to the end of enterocin substance application. This could lead to consideration that longer application of CCM7420 strain did not influence more the *Eimeria* sp. oocysts counts in rabbits.

While probiotics are widely used in animals because they improve the growth performance, productivity, health status and stimulate the immunity, studies concerning the protective effect against *Eimeria* sp. are still limited and focused on mainly the avian coccidiosis [65,66]. In rabbits, natural alternatives—prebiotics and herbal extracts—to coccidiostats have been studied [52,67,68,69]. To the best of our knowledge, only several works demonstrated the anticoccidial effect of beneficial microbes and/or probiotics as well as their antimicrobial products bacteriocins in rabbits [24,25,28,52,53,55,70]. The in vitro effect of four probiotic/bacteriocin-producing strains towards poultry *Eimeria* sp. oocysts was also documented by Strompfová et al. [71] and no differences in the reductive effect of bacteriocin-producing and non-producing strains (*p* < 0.05) were found in this experiment. The in vivo administration of bacteriocin-producing and probiotic strains decreased *Eimeria* sp. oocysts in rabbits. Outgoing from these results, we supposed that anticoccidial effect could be done due to the lactic acid production or by the effect of bacteriocins produced by the mentioned strains. Moreover, the potential protective effect of the CCM7420 strain against zoonotic *Trichinella spiralis* infection was also investigated in the framework of a new therapeutic strategy aimed at using probiotics to control parasitic zoonoses [72], when the authors demonstrated the reduced intensity of *T. spiralis* infection and female fecundity ex vivo and in vitro (about 60%) throughout the CCM7420 administration.

### 3.4. Effect on Serum Biochemistry

The measurement of biochemical parameters is data mostly used for diagnostic investigations and presents a useful way for controlling the health of animals. However, in some cases, several “components” of the host biochemistry are less specific because of the reparation/compensation ability of healthy tissues/organs and/or metabolic processes.

The tested serum parameters were in the range of normal values defined for these parameters in previous studies with rabbits [73,74,75], although there are differences in physiological or reference ranges in rabbit serum. During the *E. faecium* CCM7420 application, increased (even though not significantly) concentration of the total protein (TP) was noted in most experiments, and remained stable or higher also three weeks after the strain ceasing (cessation). The highest increase in TP was measured through the fresh CCM7420 culture administration in rabbits (Table 3) [24,25,26], while the lyophilized (freeze dried form resolved in water and mixed in pellets did not affect the TP level [60]. Application of enterocin Ent7420 as well as the combinative application of CCM7420 with the *E. senticosus* extract also improved the TP concentration [25,26]; similarly as was reported by Fathi et al. [33] and Kalma et al. [51] who observed a slight increase in serum TP after probiotic supplementation in rabbits. The increased level of TP could be explained by better resorption and utilization from the gut; this finding could be also confirmed by higher ADWG. On the other hand, application of non-autochthonous probiotic strains *E. faecium* CCM4231 and AL41 as well as their enterocins did not affect the TP in blood serum [21,29,38]. Blood glucose is an important source of energy for many cells and this is a parameter of the balance between glucose source/availability and utilization. Similarly to TP, higher glucose content was observed during fresh (both 2 and 3 weeks) and lyophilized CCM7420 culture mixed in feed as well as during Ent7420 application. The increased glucose level can be explained by conversion of lactic acid to pyruvate through the gluconeogenesis in the liver. Oppositely, reduced glucose level was observed in the case of the rehydrated–lyophilized CCM7420 strain and its combination with *E. senticosus* (*p* < 0.01). It could be that the glucose concentration was reduced by increased H^+^ concentration due to higher organic acid values in the cecum content, which inhibited gluconeogenesis [76]. The increased H^+^ (lactate accumulation) in the organism first stimulates physicochemical mineral dissolution by increasing the osteoclast and osteoblast activity (bone resorption) and mostly the Ca^2+^ and Mg^2+^ reabsorption in renal tubules for pH neutralization; it is usually confirmed also with higher serum calcium levels. This hypothesis was confirmed also by us, when the slight increase of calcium content was noted during three weeks CCM7420 fresh culture and its Ent7420 application. The rabbits’ blood calcium levels fluctuate widely, dependent upon the level of calcium in their diet and the intestinal absorption as well [77]; this is a difference in the calcium metabolism from other mammals. While Lauková et al. [21] and Szabóová et al. [35] described no influence of probiotic strains on serum glucose, triglycerides and calcium levels, Fathi et al. [33] presented numerical increase of triglycerides associated with dietary probiotic treatment in rabbits, similarly to our achievements [25]. The hypocholesterolemic effect of probiotics in rabbits has been already presented [30,34,51]; surprisingly, our results did not confirm those findings. Moreover, increased/higher cholesterol levels (however, without significant changes) were measured during fresh culture CCM7420 and its combinative administration with *E. senticosus* [26,55] in rabbits; the detected levels were still within the physiological norm.

Exogenous factors such as manipulation, nutritional, weather and temperature changes (mainly hot environmental conditions) often induce physiological oxidative stress, which is avoided by the host’s defense system. The host’s reaction to stress can be marked mostly by the glutathione-peroxidase (GPx) enzyme activity in blood. In addition, there were no significant differences in GPx activity in blood among experimental groups whereas they were differently affected by CCM7420 (Table 3). Application of CCM7420 strain did not disturb the GPx level; while Ent7420 has a reducing effect on GPx during its application (day 21; *p* < 0.05; [25]). Comparing our previous experimental applications of probiotic strains and their enterocins in rabbits, we suppose that enterocins were more active to protect the host organism; our assumptions were confirmed by reduced GPx levels during Ent7420 and EntM application [25,36]. Outgoing from results we suppose that application of CCM7420 and its bacteriocin Ent7420 did not evoke oxidative stress in the rabbits, similarly to other probiotic strains or enterocins administration [21,29,38].

### 3.5. Effect on Organic Acids

In rabbits, approximately 40% of digested organic matter of the feed is digested in the caeco-colic segment; so the caecum and the proximal colon are the primary fermenters [78]. The digestion process of nutrients continues in the small intestine by the digestive enzymes of the host, but some components, e.g., plant cell walls and fibers (mainly lignins, cellulose, hemicellulose, pectins) are hydrolyzed by bacterial enzymes into soluble smaller compounds and fermented into the end products: volatile fatty acids (VFA: acetic, propionic and butyric acid), ammonia, intermediary metabolites (lactic, succinic and formic acid) and gas (CO_2_, CH_4_ and H_2_; [79]). The stable microbial fermentation is essential for rabbit health, and only small dietary and environmental changes can lead to increased morbidity and/or mortality via microbial dysbiosis and digestive disturbances. Natural substances applications could prevent those disturbances [12,13,14]. The concentrations of VFA are usually measured in the cecal content of rabbits [21,28,29,30,38,78]; in our first experiment we decided to follow the VFA and organic acids concentrations in feces (it was a model experiment with a low number of rabbits in the experimental groups). Application of *E. faecium* CCM7420 to rabbits led to an increase of fecal levels of acetic acid (*p* < 0.001) compared to control animals [24], while other tested organic acids (butyric, succinic and lactic acids) were unaffected with the CCM7420 treatment. Similarly to our results, application of other probiotics (*E. faecium* CCM4231; [28] or bacteriocins (EntM and nisin; [21,38]) in rabbits did not influence the molar proportion of VFA in caecum, while the total VFA production and cecal fermentative activity was increased after *L. plantarum* spray application [29]. Concluded from these results—enhanced enzymatic activity and organic/fatty acid production, better feed conversion ratio and improved jejunal morphology (data shown below) during the CCM7420 strain application, we hypothesized a positive correlation between weight gain and cecal fermentation, improved gut functionality (jejunal morphology) and nutrient absorption. Despite many reports presenting the beneficial results during natural substance application in rabbits, there is a need to extend the existing knowledge and find new possibilities to improve the cecal fermentation and rabbit gut health.

### 3.6. Effect on Immunity and Jejunal Morphometry

Knowledge of the immune response and homeostasis in farm animals represents important information to protect animals from especially bacteria-derived diseases, to improve their health and productivity. The overall organization of the rabbit’s digestive immune—lymphoid system is mostly similar to other species, but at the same time it is very special. There are two additional structures identified only in this species, the sacculus rotundus and the vermiform appendix (a place of lymphoid cells differentiation and maturation); they generally act synergistically. The gut microbiota contribute to intestinal homeostasis via inducing the intestinal immune cells and also influence the systemic host immunity. The probiotic consumption shows a beneficial effect in several ways, including intestinal microbiota balance and ability to modulate host innate and specific immune response. Their effect on non-specific immunity was reported as enhanced phagocytosis of pathogenic bacteria and modified cytokine production [80], while the specific way is usually followed through immunoglobulins testing. Nevertheless, the effect of probiotic administration on the immune system of rabbits has been reported on a limited scale [61,81]. Our studies with *E. faecium* CCM7420 alone and in combination with *Eleutherococcus senticosus* demonstrated significant increases in total phagocytic activity (PA) of leukocytes and PA of neutrophils at the end of the treatment period (21 days) and also after three weeks of the ost-treatment period (42 days; *p* < 0.0001; Table 3; [26]). The freeze-dried CCM7420 strain has not influenced the PA in rabbits during its application, either resolved in water or composed in pellets. However, the prolonged effect of CCM7420 strain rehydrated in water was observed at the end of the experiment (42 days; *p* < 0.0001; [55]). Another rabbit studies with non-autochthonous strains *E. faecium* CCM4231 (ruminal isolate) and AL41 (CCM8558; isolate from animal waste) also showed stimulation of non-specific immune reaction in rabbits; the significant increase of PA (*p* < 0.001) was noted in both experiments during the treatment and increased several weeks after the strain’s cessation. Fathi et al. [33] also represented improved cell-mediated immunity adding 400 g probiotic/t feed in rabbits´ diet. Contrary to results reported above, Wang et al. [57] noted no probiotic influence on the number of mast cells in duodenum and jejunum, but increased the number of mast cells in caecum and also, increased IgG and IgM in serum. During enterocin Ent7420 administration in rabbits, the prolonged immuno-stimulative effect was observed, which was demonstrated by a significant increase of PA in the experimental group (*p* < 0.05) compared also to the control data [82]. The same immunomoderate influence was noted during experimental applications of enterocins produced by strains CCM4231 and AL41 (Ent4231 and EntM/EntAL41; [21,28]) as well as in the case of nisin feed inclusion in rabbits [38]. We expected that enterocins are able to stimulate the immune system through the gut microbiota modulation on behalf of lactic acid bacteria (LAB) and via supporting/improving the GALT by the stimulation of the IgA system. This fact could explain the “timeshift“ of the enterocins influence compared to the probiotic strain application, since probiotic bacteria begins to act and multiply immediately, while bacteriocins firstly modulate the environment on behalf of beneficial microbes (LAB) inhibiting other bacterial species in the gut. On the other hand, the prolonged or maintaining immunostimulative effect (even at 21 days after ceasing the administration of enterocins) should be also explained by the adopting of animals on them. More studies testing the effect of the CCM7420 strain as well as its enterocin Ent7420 on the immune response, containing PA and other parameters of innate immunity in rabbits are necessary. Moreover, we would like to focus also on the intestinal immunity, mainly on the IgA level.

It is well known, that probiotic supplementation improves not only the growth rate and enhances the efficiency of feed conversion but also may positively influence the health status via enhancing gut health in rabbits [61]. Gut health, including microbial and immunological stability, is often influenced by exogenous factors (dietary changes, stress from manipulation, transfer, climate changes, etc.), mainly around the weaning period. Therefore, the alternative strategies are required to improve the animal’s health. The stable or improved intestinal environment (gut microflora, mucosal immunity, epithelial morphology and function) directly influences the health status and growth performance of animals due to better nutrient absorption in the gut [61,83]. The effect of natural feed additives, including probiotics on the intestinal histomorphology in animals is often presented, mostly in chicken and pigs. However, studies reported changes of morphometric parameters in the small intestine of rabbits during probiotic application are limited [31,82,83,84]. Our results showed that the surface area and villi height: crypt depth (VH:CD) ratio also increased throughout the *E. faecium* CCM7420 strain administration [83]. These results could support the hypothesis about higher weight gain, better feed conversion and nutrient utilization due to enlargement of the absorption surface and improved morphometry parameters. Similarly to these results, increased surface area, VH, VH:CD and decreased CD was observed after Ent2019 addition in rabbits. To the best of our current knowledge, our team is the first that deals with experiments regarding the morphological changes in rabbits during enterocins administration (Ent2019 and EntM; [80,85]). This knowledge leads us to a more detailed study of these physiological changes in the rabbit’s digestive tract, which will not only allow us to extend our knowledge, but also gain new information to understand the complexity of physiological, microbiological and immunological processes in the host organism.

### 3.7. Effect on Meat—Nutrient Content and Physicochemical Properties

Rabbit meat is greatly valued for its high nutritional and dietary quality, especially its low amounts of cholesterol, fat and sodium and high content of polyunsaturated fatty acids (PUFA), potassium, phosphorus and magnesium [85,86]; for these reasons it is recommended mainly for children, pregnant women and patients with cardiovascular illnesses. Previously there have been many studies/reviews concerning rabbit meat, including its production, quality, physicochemical properties and composition [1,2]. A lot of them dealing with the effects of dietary supplementation with functional compounds, mainly probiotics, prebiotics, fatty and organic acids, vitamins, selenium and antioxidants and their combinations on rabbit carcass quality [12,13,14,31,33,40,87,88,89]. In general, there is a lack of studies testing the beneficial microbiota and/or their antimicrobial substances (bacteriocins) on rabbit meat quality and composition, including fatty acids, amino acids and minerals [31,41,42,43,44,45]. Moreover, to the best of our knowledge we presented the first reports about the effect of beneficial/probiotic strains as well as bacteriocins on the minerals and amino acids of rabbit meat [39,40,42] and also the effect of bacteriocins on the fatty acid content of rabbit meat [41,42,43]. All presented results concerning the physicochemical properties—pH, protein, fat, ash and water content, energy value, lightness and color of rabbit meat—showed that probiotic administration in rabbits had no negative effect on the rabbit meat quality. On the other hand, Fathi et al. [33] presented a significant effect of probiotic supplementation on moisture, dry matter, organic matter and ash content. Only limited information about rabbit minerals are available, in response to natural feed additives. Our result showed increased iron content (*p* = 0.0011) in treated groups with freeze-dried CCM7420 strain (both, rehydrated in water and enriched in feed pellets), in contrary to other findings with reduced iron content during microbial fermented feed utilization with *Lactobacillus plantarum* and *Pediococcus acidilactici* [90] and after enterocin administration [42]. Although the pH of the luminal content was not measured, we hypothesized a more acidic environment in the gut due to the previous results of higher lactic acid production and lower pH in the caecum during CCM7420 strain administration in rabbits [48]. This acidic environment can enhance the ionization of minerals, which in turn results in passive diffusion [91] and could be one alternative explanation of the higher iron absorption from the gut. Another hypothesis could be the larger absorption surface due to enterocyte proliferation, which is confirmed by improved morphometry parameters—villus height, crypt depth and villus height:crypt depth ratio—also recorded during our previous experiments with CCM7420 administration [82] and lantibiotic–nisin application to rabbits [38]. The enlargement of the luminal surface could ensure better mineral absorption and their inclusion to rabbit meat. On the other hand, significantly decreased concentrations of copper (*p* = 0.0004) and calcium (*p* < 0.0001) were noted. Similarly to these results, the copper concentrations in rabbit meat also reduced during the enterocin M addition to rabbits, however, not significant but only numerically changes were recorded. Similarly to us, lower concentrations of copper, zinc and manganese was found by Shah et al. [90] after probiotic supplementation. Copper is an essential trace element, performing important biochemical functions; its level in rabbit meat varies widely [92,93]. Regarding the higher iron concentration, we hypothesized an iron competitive influence on the copper intestinal absorption and its lower deposition to meat. The calcium metabolism in rabbits is very unique, widely fluctuates and its intestinal absorption is very vitamin D independent, in contrast to most mammals [77]. Despite the generally known fact that probiotics can increase the organic and short chain fatty acids in caecum, which also stimulate the minerals ionization and diffusion through the intestine, we noted decreased calcium concentration, although, still in the range presented in the literature [86,92]. Inferring from achieved results, we assumed no adverse effect of the CCM7420 strain on the meat characteristics; in addition, it could enhance the mineral quality of meat and also increased its value to the functional food level.

## 4. Conclusions

The strain *E. faecium* CCM7420 in different application forms (fresh culture at concentration 1 × 10^9^ CFU/mL of cells in a dose 500 μL/animal /day applied into drinking water) was lyophilized (freeze-dried) from rehydrated water (1 × 10^9^ CFU/mL; dose 500 μL/animal/day) as well as mixed in feed and pelleted (15 g/100 kg feed), either alone and in combination with *Eleutherococcus senticosus* and its enterocin Ent7420 (50 μL/animal/day applied into drinking water) were tested in rabbits. During these experiments, the following effects of the strain and its enterocin were observed: improved average daily weight gain and feed conversion ratio, good colonization ability of the tested strain with maximum counts in the first 2–3 weeks of application, increased lactic acid bacteria and reduced coagulase-positive staphylococci including *S. aureus*, coliforms and clostridia population as well as the *Eimeria* sp. oocysts counts in the rabbit’s gut. Improved biochemical blood parameters (total proteins, glucose and triglycerides) have been noted during the CCM7420 strain application; however, the glutathione-peroxidase level was not disturbed and oxidative stress was not evoked through the additives application. Another interesting finding was the significant stimulation of blood phagocytic activity and also the improved morphometry parameters (enlargement of the absorption surface in jejunum and higher villi height: crypt depth (VH:CD) ratio). The physicochemical properties of rabbit meat were not negatively affected by the CCM7420 strain, while the meat iron content significantly increased during its application, which improved the rabbit meat quality. It could be also emphasized that knowing the probiotic properties and the ability of *Enterococcus faecium* CCM7420 to produce enterocin Ent7420 with an antimicrobial effect is of great interest mainly in the case of several disease/pathologies, such as epizootic rabbit enteropathy, which are difficult to prevent and combat because their etiology is not known and there is no vaccine. This strain is the main component of the Prorabbit probiotic preparation, which is often used in Slovak rabbit farms (at dosage 1–2 g/animal/day for 21 days as prevention and 3 g/animal/day with a therapeutic effect; resolved in water or mixed into feed). Moreover, to the best of our current knowledge, our team is the first that deals with experiments regarding the morphological changes in the jejunum of rabbits during enterocins administration; the first reports regarding the effect of beneficial/probiotic strains and bacteriocins on the mineral and amino acid concentrations as well as the effect of bacteriocins on fatty acid content of rabbit meat were also published by our team.

## Figures and Tables

**Table 1 animals-10-01188-t001:** The effect of *Enterococcus faecium* CCM7420 and its enterocin Ent7420 on the growth performance of rabbits.

Tested Zootechnical Parameters	*E. faecium* CCM7420 Strain					Enterocin (Ent) 7420
	Fresh Culture	Fresh Culture	Lyophilized Form Resolved in Water	Lyophilized Form Mixed into Pellets	Fresh Culture + *E. senticosus*	
Reference number of publication	[24]	[25]	[55]	[55]	[26]	[25]
Length of application	14 days	21 days	21 days	21 days	21 days	21 days
Number of rabbits	(*n* = 7)	(*n* = 24)	(*n* = 24)	(*n* = 24)	(*n* = 24)	(*n* = 24)
Initial live weight (35 d of age; 0 d of experiment), g	1136.0 ± 100.0	977.0 ± 97.0	1002.3 ± 162.3	1042.5 ± 315.7	1077.5 ± 102.2	963.0 ± 101.0
Intermediate live weight (49/56 d of age; 14/21 d of experiment), g	1507.0 ± 120.0	1850.0 ± 152.0	1664.4 ± 170.0	1856.9 ± 361.4	1943.3 ± 222.2	1788.0 ± 199.0
Final weight (70/77 d of age; 35/42 d of experiment), g	2325.0 ± 260.0	2622.0 ± 104.0	2206.7 ± 164.6	2319.2 ± 164.6	2723.3 ± 204.7	2431.0 ± 142.0
Average daily weight gain (g/d; increase compare to control data%)	28.00 (22.0%)	39.17 (4.8%)	38.35 (10.0%)	38.51 (10.5%)	39.19 (8.5%)	34.95 (2.2%)
Feed conversion ratio between 35 and 56 days of age (g/g)	Not tested	2.71	2.23	2.28	2.59	2.68
Feed conversion ratio between 56 and 77 days of age (g/g)	Not tested	4.67	3.41	3.42	3.97	3.87
Feed conversion ratio per kg gain	Not tested	3.47	2.82	2.85	3.28	3.22
Mortality (*n*(%) in experimental group/*n*(%) in control group)	0(0.0%)/0(0.0%)	3(12.5%)/5(20.8%)	3(12.5%)/7(29.2%)	3(12.5%)/7(29.2%)	0(0.0%)/4(16.7%)	1(0.04%)/4(16.7%)

The fresh culture of the CCM7420 strain was applied into water (at concentration of cells ×10^9^ CFU/mL; dose 500 μL/animal/day); lyophilized (freeze-dried) form rehydrated in water (×10^9^ CFU/mL; dose 500 μL/animal/day) as well as mixed in feed and pelleted (15 g/100 kg feed).

**Table 2 animals-10-01188-t002:** The effect of *Enterococcus faecium* CCM7420 and its enterocin Ent7420 on the fecal microbiota of rabbits.

	*E. faecium* CCM7420 Strain					Enterocin (Ent)7420
Tested Microorganisms	Fresh Culture	Fresh Culture	Lyophilized form Resolved in Water	Lyophilized Form Mixed into Pellets	Fresh Culture + *E. senticosus*	
Reference number of publication	[24]	[25]	[55]	[55]	[26]	[25]
Length of application	14 days	21 days	21 days	21 days	21 days	21 days
Enterococci	Increased	Increased (*p* < 0.01)	Increased	Increased	Unchanged	Increased (*p* < 0.05)
Lactic acid bacteria (LAB)	Increased (*p* < 0.01)	Unchanged	Increased	Increased	Unchanged	Decreased
Clostridia	Decreased	Decreased	Unchanged	Unchanged	Decreased (*p* < 0.05)	Unchanged
Coagulase-positive staphylococci	Decreased (*p* < 0.01)	Unchanged	Unchanged	Unchanged	Decreased	Decreased
*Staphylococcus aureus*	Unchanged	Decreased	Unchanged	Unchanged	Decreased (*p* < 0.001)	Decreased
Coliforms	Unchanged	Decreased	Unchanged	Unchanged	Decreased (*p* < 0.001)	Decreased
CCM7420	6.7 log cycle	6.7 log cycle	<1.0 log cycle	2.8 log cycle	1.1 log cycle	-

The fresh culture of the CCM7420 strain was applied into water (at concentration of cells ×10^9^ CFU/mL; dose 500 μL/animal /day); lyophilized (freeze-dried) form rehydrated in water (×10^9^ CFU/mL; dose 500 μL/animal/day) as well as mixed in feed and pelleted (15 g/100 kg feed). Statistical analysis was performed using one-way analysis of variance (ANOVA) with the post hoc Tukey test with the level of significance set at (*p* < 0.05), within experimental groups during each individual experiment.

**Table 3 animals-10-01188-t003:** The effect of *Enterococcus faecium* CCM7420 and its enterocin Ent7420 on the serum biochemistry of rabbits.

	*E. faecium* CCM7420 Strain					Enterocin (Ent)7420
Tested Blood Parameters	Fresh Culture	Fresh Culture	Lyophilized Form Resolved in Water	Lyophilized Form Mixed into Pellets	Fresh Culture + *E. senticosus*	
Reference number of publication	[24]	[25]	[55]	[55]	[26]	[25]
Length of application	14 days	21 days	21 days	21 days	21 days	21 days
Total proteins (g/L)	Increased (*p* < 0.05)	Increased	Unchanged	Unchanged	Increased	Increased (*p* < 0.05)
Total lipids (g/L)	Increased	Increased	Unchanged	Unchanged	Unchanged	Increased
Cholesterol (mmol/L)	Not tested	Not tested	Unchanged	Unchanged	Increased	Not tested
Glucose (mmol/L)	Unchanged	Increased	Increased	Decreased	Decreased	Increased
Calcium (mmol/L)	Unchanged	Increased	Unchanged	Unchanged	Not tested	Increased
Glutathione-peroxidase (GSH-Px; U/mL)	Increased	Decreased	Increased	Increased	Decreased	Decreased
Phagocytic activity (%)	Not tested	Not tested	Increased	Increased	Increased (*p* < 0.0001)	Not tested

The fresh culture of CCM7420 strain was applied into water (at concentration of cells 1 × 10^9^ CFU/mL; dose 500 μL/animal /day); lyophilized (freeze-dried) from rehydrated in water (1 × 10^9^ CFU/mL; dose 500 μL/animal/day) as well as mixed in feed and pelleted (15 g/100 kg feed). Statistical analysis was performed using one-way analysis of variance (ANOVA) with the post hoc Tukey test with the level of significance set at (*p* < 0.05), within experimental groups during each individual experiments.

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
