# Peer review of "Autochtonous Strain Enterococcus faecium EF2019(CCM7420), Its Bacteriocin and Their Beneficial Effects in Broiler Rabbits—A Review"

_animals, 2020, doi:10.3390/ani10071188_

Round 1

Reviewer 1 Report

This is an interesting review about a really interesting topic in rabbits.

The title should indicate this work is a review.

In general, I missed the inclusion of the real values of the traits studied, the data of the control groups, and the inclusion of graphs rather than tables. I wonder if a meta-analysis could be done to summarise and quantify better the effects. I would include a practical conclusion about the dose and the best form to supply these additives.

I have included my comments in the attached pdf file.

Sincerely yours,

Author Response

Response to Reviewer 1

Reviewer 2 Report

GENERAL COMMENT:

I consider this work is within the scope of “Animals”. It contains information useful in a field in which available information is of interest to improve gut health of fattening rabbits. The manuscript is well written and structured, and its revels expertise of the authors on the subject. I indicate only few suggestions to be considered by the authors to improve the manuscript. I indicate these recommendations below and in a commented PDF file I have uploaded.

SIMPLE SUMMARY:

Lines 17-19: Please review writing of the following sentence: “Control of the microbiota, prevention of mentioned digestive disturbances and improving gut health and immunity through the application of natural substances in rabbit nutrition”. It seems to be incomplete.

INTRODUCTION:

Lines 63-64. Please review format of literature quotation in the text. You have skipped reference number [5]. Probably it is FAO (2019).

Near Line 62: in this section, it could be emphasized that knowing the probiotic properties of Enterococcus faecium CCM7420 is of great interest if one takes into account that certain pathologies, such as Epizootic Rabbit Enteropathy, are difficult to prevent and combat because their etiology is not known and there are no vaccines.

SECTION 2:

Line 121: Phrases should not start with initials. Therefore, write “E.” with the entire word: “Enterococcus”.

SECTIONS 3.

Line 161: Correct typo: remove one of the "s" duplicated at “ssenticosus”.

Line 170: Other commercial products containing species strain Enterococcus faecium as a component of probiotic bacterial mix are Pro-enteric Triplex (DSM 10663/NCIMB 10415). It has been tested and results reported in El Dimerdash, M.Z., Dalia, M.H., Hanan, F.A., & Doaa, S.A. (2011). Studies on the effect of some probiotics in Rabbits. SCVMJ, 16(2), 151-168.

https://www.researchgate.net/profile/Doaa_Elhalous/publication/331907843_Studies_on_the_effect_of_some_probiotics_in_Rabbits/links/5c92b6e0a6fdccd4602e1d5f/Studies-on-the-effect-of-some-probiotics-in-Rabbits.pdf

You can refer to it in this part of the article.

Line 231, section relate to : Indicate whether the oocysts reduction after the experimental applications of E. faecium, has shown or not differences among Eimeria species.

Line 231: Renumber section as 3.3

Line 282: Renumber section as 3.4

Line 338: Renumber section as 3.5

Line 365: Renumber section as 3.6

Line 430: Renumber section as 3.7

Lines 478-480: Remove the following fragment: “This section may be divided by subheadings. It should provide a concise and precise description of the experimental results, their interpretation as well as the experimental conclusions that can be drawn.” Because it belong to the journal template.

TABLES:

Table 1, title: Write Enterococcus rather than E. because tables must be interpreted independenty of the manuscript text.

Table 1: Indicate mortality also as a percentage. The single number does not illustrate the magnitude of the effect.

Table 1: "conversion", rather than "conversation"

Table 2, title: Write Enterococcus rather than E. because tables must be interpreted independenty of the manuscript text.

Table 3, title: Write Enterococcus rather than E. because tables must be interpreted independenty of the manuscript text.

Table 3: correct typo at: “mmolúL”

CONCLUSION SECTION:

I suggest you to include in the Conclusions section a brief sentence on the recommendation of practical application of this probiotic to fattening rabbit.

REFERENCES SECTION:

Review this section to remove typos and to better adjust it to the journal style. I have indicated some of them in the annotated version of the manuscript I have uploaded.

MORE LITERATURE SOURCES POTENTIALLY RELEVANT:

You can consider checking whether any of the following papers have useful information to be considered in your manuscript. For example:

Benato, L., Hastie, P., O'Shaughnessy, P., Murray, J. A., & Meredith, A. (2014). Effects of probiotic Enterococcus faecium and Saccharomyces cerevisiae on the faecal microflora of pet rabbits. Journal of Small Animal Practice, 55(9), 442-446.

Silva, N., Igrejas, G., Figueiredo, N., Gonçalves, A., Radhouani, H., Rodrigues, J., & Poeta, P. (2010). Molecular characterization of antimicrobial resistance in enterococci and Escherichia coli isolates from European wild rabbit (Oryctolagus cuniculus). Science of the Total Environment, 408(20), 4871-4876.

Author Response

Response to Reviewer 2

Reviewer 3 Report

A review concerning a probitical alternative for antibiotics is interesting, especially to judge the efficacy on a much larger number of animals and trials. Moreover, if works concerning the working mechanism are introduced and eventually the lacks in knowledge, such a review is very valuable. 

However two main facts are important for a review: to be critical for the literature found on the topic and to try to analyse the global data so that more reliable scientific conclusions can be drawn.

The proposed review fails partly in these 2 objectives. Below some major remarks   

Comments:

Introduction: too long and too general: the first 2 paragraphs have nothing to deal with the study and can completely be deleted or reduced to 1 or 2 sentences.

Lines 151-156: The value of production performances (DWG, FCR...) are well known information to judge the efficacy and this has not to be mentioned because already longtime known and accepted. AVOID general and simplified information. Stay on the topic!! 

L156- Results:  e.g. DWG or FC: are the mentioned effects significant? Authors mention improved results but not clear if significant ??? If you compare 2 groups of animals, with the SAME treament, always you will find (small) differences but only if they are statistical significant, than you may speak about positive (or negative) effects or improved. In the different tables information about the effects is lacking.

If the cited trials does not allow to judge the effects (e.g. because of a too low number of replicates), they have to  be removed from the dataset. Another possibility is to join the data of the different trials mentioned and to make a data set on which a statistical tests can be performed.

I have only evaluated the results presented in the first 2 subchapter of chapter 3. A more critical judgement of the literature has to be done and especially a more global evaluation of the different trials mentioned. 

However,  in chapter 3 the different subchapters are mentioned several  times as 3.1 ...

Author Response

Response to Reviewer 3

Round 2

Reviewer 3 Report

Authors have done a significant effort to improve their review in order to present a more scientific and criticial review. 

In the response letter, they explained why they did not pool the results of different trails with the Enterococcus faecium EF2019(CCM7420) strain. 

Results in the different tables are now marked if differences are significant.

Based on the above mentioned reasons, now I can agree that this review as accepted for publication.